# Lost in the crowd: Imagining walking in synchrony with a crowd increases affiliation and deindividuation

**Benjamin Philip Crossey[1], Gray Atherton[2], Liam Cross[2]***

**1** School of Psychology, Queen's University Belfast, Belfast, United Kingdom, **2** Department of Psychology, Edge Hill University, Liverpool, United Kingdom

* drliamcross@gmail.com

**Data Availability Statement:** The data underlying this study are available on OSF (https://osf.io/dtcq7/).

**Funding:** The author(s) received no specific funding for this work.

## Abstract

Moving in time with others—interpersonal coordination—increases affiliation, helping behaviours and gives rise to a host of other prosocial outcomes. Recent research suggests that merely imagining coordination may lead to similar social effects. In the present study, participants were asked to imagine walking with a crowd in a coordinated (versus uncoordinated) way to explore the effects of imagined coordination on individuals' perceptions of themselves and the crowd. Imagined coordination led to greater levels of deindividuation and affiliation. That is, participants were less likely to report seeing themselves as unique individuals, instead viewing themselves as a part of a group (deindividuation) and more likely to report a sense of emotional closeness (affiliation) with the imagined group. Deindividuation partially mediated the effect of imagined coordination on affiliation. This work establishes that imagined synchrony can be employed online to foster prosocial attitudes towards groups of people, and that a process of deindividuation might mediate this effect.

## Introduction

Deindividuation and the perceptions of responsibility linked with membership in a crowd are among the most long-standing areas of interest in social psychology and include some of its most well-established findings (e.g., [1, 2]). Indeed, many studies in social psychology have paid particular attention to the way that people, once absorbed in a crowd, act less with reference to the personal or individual self and more in line with the collective [3]; see also: [4]) In the present investigation we reveal that the deindividuating effects of crowd membership extend beyond physical crowds to include membership in imagined, unspecified crowds; that the motoric aspects of the crowd are important, and that some of the effects that are usually attributed to synchronous movement can be obtained via imagination alone.

Coordinated interpersonal movement has been shown to elicit a range of social consequences that can arise irrespective of movement type (for reviews see, [5–8]). Though traditionally more speculative (e.g., [4]), the previous decade has seen an increase in the number of experimental investigations into the prosocial effects of motoric coordination. In a seminal study, participants who tapped in synchrony with an experimenter later rated the

**Competing interests:** The authors have declared that no competing interests exist.

experimenter as more likeable than those who tapped asynchronously with them [9]. This increase in affiliation following coordination has been demonstrated in several studies [10–12] and some research indicates that the increased level of affiliation that is evoked by synchronous movement may also lead to more prosocial behaviours. An example of this is that synchronous walking, or cup waving and singing, results in participants making more cooperative choices in subsequent economic games, even if such choices come at a real financial cost to them [13]. Other research has shown that after moving synchronously together, participants are more likely to help their co-actors, even some time after the initial coordination task [14], as well as to conform and obey their requests [15, 16], and show better memories for co-actors [17–19].

Interestingly, some findings suggest that deindividuation—a known element of crowd behaviour—may itself play a meditational role in these social outcomes [5]. In other words, although deindividuation usually describes the phenomenon by which membership of a group reduces individuals' sense of individuality, personal responsibility, and self-awareness (e.g., [20]), it may also give rise to the greater levels of coherence, affiliation and prosociality for fellow group members that follows interpersonal coordination. Consistent with this, it has been shown that moving in synchrony causes individuals to be more likely to think of themselves as part of the collective rather than as separate individuals [10]. Similarly, another study has demonstrated that after interpersonal coordination, individuals perceive themselves in more interdependent terms [21]. This effect is not only present in self-reports but can also be detected by external observers [22]. Moreover, participants who see themselves in more interdependent (i.e., relational) rather than individualistic terms, think and act in more prosocial ways and have been shown, for example, to be more likely to cooperate [23–25].

Consistent with the notion that coordinated movement may lead to greater prosociality as mediated by deindividuation levels, it has been reported that two experiments showing that synchrony not only increases the odds that a participant will be willing to help others, but that their likelihood of helping is directly related to their reported levels of deindividuation [14]. That is, the greater the level of deindividuation evoked by synchronous movement, the more likely participants are to offer help. The present experiment investigated whether deindividuation mediates such outcomes.

A key issue with the idea that synchrony increases deindividuation which in turn mediates affiliation and prosociality, is that both affiliation and prosociality were initially understood to arise in group movement as a result of endorphin levels. Although, it has been argued that by engaging in synchronised drills soldiers may develop a sense of 'we-ness', and is also suggested that the 'fellow-feeling' that such drills give rise to are the result of the interaction between endorphins themselves arising from the physical components of movements, and experience [4]. In other words, endorphins are likely largely responsible for the affiliation and deindividuation levels evoked by group coordination (see [4]). Although compelling, the notion that the social effects of synchrony arise as a consequence of an interaction between the endorphins that are released as a consequence of gross motor movement and the experience of moving synchronously with others is not fully supported by the literature. In fact, one of the first experimental studies to show a direct link between synchrony and prosociality found these effects to be present following synchronous fine muscle movements (e.g. finger tapping [9]), which do not result in endorphin release, unlike strenuous gross motor movements (for example see [26]). In part, the present experiment set out to investigate whether the affiliative effects of synchrony can be obtained following gross motor movements (e.g., walking, marching, etc.) without the presence of endorphins, indeed, without any actual movement at all. To achieve this, we asked participants to merely imagine a physical movement interaction rather than to actually make the movements, in order to determine whether comparable social outcomes can be evoked without any possibility of muscular or other physical movement effects. If similar

outcomes are evoked then clearly the movements themselves are not essential to the social outcomes of coordinated movement.

Recent research suggests that similar effects should be elicited following imagined rather than physical/real movements because many of the social and cognitive, non-physical aspects of coordinated movements can be activated by imagination alone. Imagined actions have been shown to recruit similar neural areas and to elicit similar physiological responses as actual performance (e.g., [27–32]). For example, one study found that people take the relevant aspects of an *imagined* co-actors' actions into account much as they do when coordinating with them in real life [33]. That is, they integrate simulations of the imagined co-actor's part of the joint action with their own. Moreover, just as practice improves performance, imagined motor tasks improve performance for the relevant action across a range of domains (e.g., [34–41]). As for prosociality, imagining being part of a crowd can result in effects that often mirror the effects of real crowd membership, such as the bystander effect [42]. Researchers have revealed that imagining a positive verbal interaction improves relationships between people similar to actual positive contact [43–46]. There is also evidence that imagining walking in synchrony can foster affiliation towards small groups of people one has previously met [47] and influence attitudes towards outgroup members just as physical synchrony does [48, 49].

Imagination paradigms have a number of advantages over physical paradigms that can be used to circumvent some of the limitations of physical research. The first is that they can be used to test effects of interactions with entirely unspecified populations. A participant can be asked to imagine an interaction with a crowd of people without there being any need to outline the specific details of the crowd. This is contrastable with research involving physical crowds in which the real crowd members will necessarily have physical characteristics that may be difficult to control for or partial out in subsequent analyses. For instance, a person may be asked to imagine synchronising with a large crowd of other people who are all walking in time with each. Achieving this in a lab environment would be more difficult as the group size increases. (We would like to stress: we do not suggest that imagination paradigms should or can take the place of physical paradigms but rather that they can be used to support their findings or act as a further source of comparison). In the case of synchrony research, even when bringing a group of strangers into the lab for a group movement experiment involving relatively small groups of people (e.g., dyads or triads) the participants involved will usually have a shared goal (i.e., to do an experiment and to get paid for it or to gain course credits) and shared characteristics (they will likely be of a similar age, the same nationality and region, be students of the same university etc, and very often study the same discipline). These extraneous factors may interact with the experience of synchronous movement. However, if alternative manipulations including imagination-based paradigms lead to the same social outcomes as 'real' interpersonal synchrony then it is less probable that such extraneous variables will play a role. In other words, an imagined coordination paradigm not only controls for the possible interaction between endorphins and experience (e.g., [4]) but other extraneous interactions such as shared goals. What's more, much is to be gained by investigating the prosocial effects of imagined synchrony in and of itself. If affiliation can be increased merely by imagining a synchronous interaction then imagined synchrony paradigms greatly broaden the availability of interpersonal synchrony to applied research.

The present experiment extends previous research and addresses existing confounds in the literature by testing whether the prosocial effects of coordinated vs uncoordinated walking can be elicited in response to imagined, *unspecified* individuals. In a paradigm presented via an online platform, participants were asked to imagine walking either synchronously or asynchronously with a crowd. No description of the characteristics of this crowd was provided. In order to better understand the relationship between coordination, deindividuation and

affiliation, this task was followed by items assessing affiliation and deindividuation. It has been discussed that 'fellow-feeling' and the process of becoming one with others, arguing that the interaction between endorphins and experience may account for the increased level of affiliation evoked by behavioural synchrony [4]. However, besides the possible effect of endorphins, it remains unclear whether deindividuation itself plays a specifically meditational role in the synchrony-affiliation relationship. Previous literature suggests that it does—the present experiment tested this explicitly. If deindividuation accounts for the affiliative outcomes observed following synchrony, we would expect it to do so for imagined synchrony, that is, without the need for physical movement, particularly given that imagined physical interactions recruit much of the same neural substrate as real physical movement. Hence, we investigated whether the effects of physical synchrony can be elicited in the absence of the potential confounds of motor movement paradigms themselves.

We hypothesised that participants who imagined walking in synchrony with a crowd would i) report a greater level of affiliation with the crowd, and ii) report a greater sense of deindividuation than those who had imagined walking asynchronously with the crowd, and that iii) deindividuation levels would mediate imagined synchrony's effect on affiliation.

## Method

### Design, materials & procedure

The study employed a single factor (imagined synchrony versus imagined asynchrony) between participants design. Participants from the United Kingdom and North America were recruited from the Universities of Buckingham, Leeds Beckett, and Houston, completing the questionnaire online in response to email requests and advertisements posted on Sona in exchange for course credit.

This study was conducted online via the survey platform, Qualtrics. Participants were instructed to complete the survey on a computer or laptop rather than on a handheld device and in a quiet place where they would not be interrupted. Individuals were randomly assigned to one of two conditions (imagined synchrony versus imagined asynchrony) and asked to close their eyes and spend two minutes imagining walking with a large crowd, presented with the following instructions:

**Imagined synchrony.** *Imagine the experience of moving in a large crowd. You are all walking in time with each other. Your feet connect with the pavement at the same time. You can hear the sound of everyone's footsteps as you move in time together.*

**Imagined asynchrony.** *Imagine the experience of moving in a large crowd. You are all walking out of time with each other. Your feet connect with the pavement at different times. You can hear the sound of everyone's footsteps as you move out of time together.*

Since the experiment was conducted online rather than in a controlled laboratory setting several checks were included which sought to determine the quality of the environment and the imagined interaction in which participants had taken part. These assessed the quiescence of the environment in which participants had undertaken the experiment. In order to ensure that all participants spent at least 90 seconds on the imagination task, time spent on this page of the survey was recorded. Anyone who progressed through this page before 90 seconds had elapsed was excused from the remainder of the survey and directed to a debrief page. Participants also responded to three items targeting whether they had been *interrupted* during the task (*no/yes*), how *persistent* (*NA* versus a scale ranging from 1 (*brief*) to 7 (*continuous*)), and how *disruptive* (*NA* versus a scale ranging from 1 (*brief*) to 7 (*continuous*)) the disturbance had been if present. Those who reported being interrupted during the task were excused from the remainder of the survey and directed to a debrief page.

Social affiliation was measured using eight items adapted from previous studies [47]; and [50]. These questions measured participants perceived *closeness*, *connectedness*, *liking*, *trust*, *rapport*, and *similarity* with the crowd, along with the degree to which respondents would *wish to see them again* and *get to know them*. All items were responded to on seven-point scales ranging from 1 (*not at all*) to 7 (*very*). Deindividuation was assessed via three items from previous research ([47] originally adapted from [51]). These were presented on seven-point scales as follows: Respondents reported the degree to which they saw themselves as (i) *an individual* or *a group member* (1–7, with 1 representing *individual* and 7 representing *group member*), (ii) *a group member* (1–7), and (iii) *a unique individual* (1–7; reverse-scored). Item order was randomised.

Finally, basic demographic information was requested (i.e., *age* and *sex/gender*) along with a series of questions assessing the degree to which they felt the crowd had been *tightly coordinated* and *in sync*. Responses were scored on a scale ranging from 1 (*not at all*) to 7 (*very*). A further three items assessed how *vivid* and *difficult* the imagination task had seemed, and how *successful* participants felt they had been at imagining group walking situation outlined in the vignette. Responses were made on seven-point scales ranging from 1 (*not very*) to 7 (*very*).

## Participants & ethics

A total of 464 people (373 female; $M_{age}$ = 22.52 years, $SD_{age}$ = 5.54 years) completed the survey and passed the check questions described above. This included 218 people in the Asynchrony condition and 246 in the Synchrony condition. Sample size was not predetermined at the design phase; the study aimed to recruit as many participants as possible within a 1 month time frame. Post hoc power analysis using G power [52] confirmed that the final sample had a .996 probability of detecting medium effect sizes ($d$ = .05; see also [48] and [47]). The experiment was approved by the University of Buckingham Psychology Ethics Review Board and the University of Houston's Institutional Review Board. Anonymised data is available online (https://osf.io/dtcq7/).

## Results

Parametric assumptions were first checked and where measures were found to deviate significantly from normality, non-parametric statistics have been reported (see Table 1). Principal components analysis was also undertaken for each of our scales to ensure they could be reliably compiled into composite scores. These analyses can be found in the S1 File.

The degree to which participants thought that the imagined crowd had been *coordinated* and *in-sync* were averaged to create an index of *imagined synchrony* ($r_s$ = .70). Participants in the synchronous condition (*Mdn* = 5.50, *IQR* = 3.00) reported greater levels of imagined

**Table 1. Shapiro-Wilk tests for each measure as a function of condition.**

|  | Imagined Synchrony | Imagined Asynchrony |
|---|---|---|
| Imagined Synchronousness | .897** | .952** |
| Task Difficulty | .961** | .971** |
| Affiliation | .991 | .987* |
| Deindividuation | .975** | .970** |

Note.
* $p < .05$,
** $p < .001$.

synchrony than those in the asynchrony condition (*Mdn* = 3.50, *IQR* = 3.00), *U* = 12089.50, *p* < .001, *r* = .48, suggesting that our manipulation resulted in different levels of imagined synchrony in line with the vignettes presented for each condition.

To create an index of *task difficulty*, the item assessing the task's *vividness* was combined with its purported *difficulty* (reverse-scored) and the degree to which participants felt they had been *successful* at imagining the scenario. This was not affected by condition, *U* = 24950.500, *p* = .194, *r* = .06. In other words, synchrony (*Mdn* = 2.67, *IQR* = 2.00) and asynchrony (*Mdn* = 3.00, *IQR* = 1.67) were similarly easy to imagine, and participants felt they were similarly successful at doing so.

The extent to which participants reported a sense of *closeness*, *connectedness*, *liking*, *trust*, *rapport*, and *similarity* with the crowd, along with the degree to which they *would wish to see them again*, and *get to know them*, were combined to create a measure of affiliation. Participants who imagined walking with a synchronous crowd reported greater levels of affiliation (*Mdn* = 3.89, *IQR* = 1.75) with the crowd than those who imagined walking with an asynchronous (*Mdn* = 3.38, *IQR* = 1.75) crowd (*U* = 21389.50, *p* < .001, *r* = .17).

The measure of deindividuation was constructed by combining items measuring whether participants identified as an individual (reverse scored), a *group member*, and the *individual-group member* rating scale. Imagined synchrony (*Mdn* = 3.33, *IQR* = 1.67) led to significantly greater levels of deindividuation than imagined asynchrony (*Mdn* = 3.00, *IQR* = 1.75), *U* = 21282.00, *p* < .001, *r* = .18.

Whether imagined coordination's effect on affiliation was mediated by deindividuation was evaluated using model 4 of the PROCESS macro available in SPSS (Hayes, 2013). Bootstrapped estimates (normality not assumed) of 95% confidence intervals (10,000 iterations) revealed that condition influenced both affiliation $F(1, 462) = 14.49$, $p < .001$, $R^2 = .03$, and deindividuation levels, $F(1, 462) = 17.53$, $p < .001$, $R^2 = .04$. Moreover, controlling for the influence of deindividuation on affiliation ($b = .39$, $p < .001$), reduced the effect of condition on outcome (see also Fig 1). Hence, we found evidence that deindividuation partially accounts for the effect of imagined synchrony on affiliation.

## Discussion

This study investigated whether imagining walking in synchrony with a crowd increases feelings of deindividuation and affiliation relative to imagining asynchronous walking. It also sought to determine whether deindividuation levels mediate the affiliative outcomes of imagined coordination. In line with our hypotheses, our results showed that individuals who imagined walking in synchrony reported greater levels of deindividuation and affiliation than those who imagined walking asynchronously. Moreover, the deindividuation levels evoked by imagined coordination mediated its effects on affiliation.

A recent review of synchrony's prosocial effects [5] highlights how the wide-ranging social effects of synchrony might result from individuals re-categorising themselves in group terms. The mediating effect of deindividuation on the imagined synchrony-affiliation relationship that we have uncovered supports this argument (see also that moving in time with others allows people to shift from a 'me' to a 'we' mindset [53] in turn leading to closer ties with group members).

However, it's important to point out that even though deindividuation may affect affiliation, and affiliation increases pro-sociality, deindividuation does not necessarily always lead to prosocial outcomes. For instance, it was found that the deindividuating effects of synchrony can lead to 'destructive obedience', tested via a novel bug-killing paradigm [15]. This experiment showed that participants were more likely to kill bugs at the behest of their co-actors

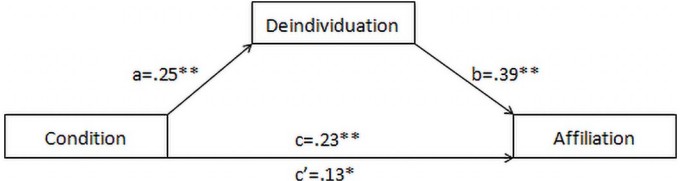

**Fig 1. Effect of imagined movement condition on affiliation.** $^*p < .05$, $^{**} p < .001$.

when they had engaged in synchronous co-action than participants who had not. Other researchers have reported anecdotal evidence that synchrony can lead to conformity and 'group think'. For instance, Herrera [54] reports on the deindividuating effects of chanting during Argentinian soccer matches through which it seems to become more permissible to take on the persona of the crowd and engage in violent and anti-social interactions. That is, deindividuation promotes feelings of affiliation with the ingroup while often also resulting in feelings of antipathy towards (and antisocial behaviour directed at) outgroup members. Our experiment demonstrates that deindividuation likely plays a critical role in affiliating the individual with the group since it changes self-categorisation from a 'me' to part of the 'we' and so bonds the individual with coordinated co-actors. Comparing the degree to which imagined and physical coordination evoke anti-social outcomes arising from deindividuation will help to determine whether these tasks give rise to shared or distinct cognitive processes.

A number of experiments have shown that imagined task performance may enhance real-life performance (e.g., [34–41]), and recruits similar neural regions and physiological responses [27–32] as physical task enactment. Other work has found that imagining a positive interaction can lead to improvements in a relationship akin to real-life positive interactions [43–46]. Similarly, our results suggest that the social effects of imagined synchronous movement mirror the effects previously found for physical synchrony. Hence our findings are consistent with simulation theory [31], which suggests that for the purposes of motor planning and execution, individuals mentally enact the given movement thus subjecting themselves to the physical effects of the action 'in vivo.'

This is the first research to show that imagining walking in synchrony (versus asynchrony) can lead to greater affiliation with an unspecified group of people. It builds on previous findings, confirming that mentally simulated synchronous walking fosters affiliation, similar to engagement in actual (i.e., physical) synchrony [47]. Our findings also lend support to the idea that this outcome is itself mediated by a shift in individuals' self-construal, leading them to perceive themselves in less individualised and more group terms. Moreover, our results suggest that individuals can experience the affiliative effects of synchronous walking online without having to be physically present with their imagined co-actors.

It has been demonstrated that individuals from specific social groups can influence their perceptions of outgroup members by simply imagining verbal interactions with them [43]. Similarly, our study suggests that imagining synchronous walking increases affiliation between self and others. However, as with intergroup verbal contact, there are several barriers that often prevent individuals from engaging in nonverbal (e.g., motor) interaction, such as isolation and physical impairment. Our research indicates that at least some of the benefits of physical interaction might be possible without a requirement for physical contact. Further research will be needed to determine the limitations of imagination paradigms in delivering some of these benefits. Nonetheless, our findings suggest that individuals need only imagine synchronous group movement in order to experience the affiliative effects.

This research sought to examine the effects of imagined synchrony with unspecified others, i.e. who are neither in-group nor outgroup members. A clear advantage of imagination paradigms is that only the relevant aspects of imagined co-actors' behaviours need be provided. This reduces the risks posed by extraneous factors (e.g., from co-actors' physical appearance). However, in some respects this begs the question: What characteristics do participants imagine? Future research may wish examine this question with a view to determining which factors are more likely to be imagined when no details about the characteristics of the imagined crowd are provided.

In order to further align our findings with contemporary physical social synchrony research, future investigations may consider testing the effects of imagined synchrony with specified others. For example, recent work suggests that actual [55] and imagined coordination [49] affect relationships across group boundaries. Indeed, it is suggested that both actual and imagined coordinated walking might influence self-other perspectives even for co-actors who are members of outgroups [48]. Determining whether the same effects hold true for imagined synchrony as seem to be present following physical synchrony would provide further evidence that imagined synchrony recruits the same neural regions as actual synchrony.

To expand on our findings, future research may wish to focus on whether imagined synchrony interventions can be used to improve the sense of social connectedness for individuals experiencing loneliness and social isolation, as well as those who are unable to physically coordinate with others. The effects of imagined synchrony and the value of imagined synchrony paradigms should also be compared with the prevailing form of imagined social intervention, imagined contact (i.e., imagined verbal interaction; see above), in which individuals imagine having a positive (e.g., pleasant) interaction with outgroup members. Comparing the outcomes of imagined coordination and imagined contact would help us better understand the relative effectiveness and degree of overlap in between these imagination tasks.

This study has demonstrated that imagined synchrony can lead to increased levels of affiliation with an imagined crowd. Our results have added to previous research by revealing that deindividuation mediates the influence of imagined coordination on affiliation. In reproducing the affiliative effects of physical coordination, our research has also demonstrated the feasibility of conducting studies of this nature online, outside of the lab.

## Supporting information

**S1 File.**
(DOCX)

## Acknowledgments

The authors would like to thank Chloe Lannen for her help in editing the final manuscript.

## Author Contributions

**Conceptualization:** Benjamin Philip Crossey, Liam Cross.

**Data curation:** Benjamin Philip Crossey, Gray Atherton, Liam Cross.

**Formal analysis:** Benjamin Philip Crossey, Liam Cross.

**Investigation:** Benjamin Philip Crossey, Gray Atherton, Liam Cross.

**Methodology:** Benjamin Philip Crossey, Liam Cross.

**Project administration:** Benjamin Philip Crossey, Gray Atherton, Liam Cross.

**Resources:** Liam Cross.

**Software:** Benjamin Philip Crossey, Liam Cross.

**Validation:** Liam Cross.

**Visualization:** Benjamin Philip Crossey, Liam Cross.

**Writing – original draft:** Benjamin Philip Crossey, Gray Atherton, Liam Cross.

**Writing – review & editing:** Benjamin Philip Crossey, Gray Atherton, Liam Cross.

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
