## [Decision Letter · Decision Letter 0]

13 Apr 2021

PONE-D-21-02076

Lost in the crowd: Imagining walking in synchrony with a crowd increases affiliation and deindividuation

PLOS ONE

Dear Dr. Cross,

Thank you for submitting your manuscript to PLOS ONE. After careful consideration, we feel that it has merit but does not fully meet PLOS ONE’s publication criteria as it currently stands. Therefore, we invite you to submit a revised version of the manuscript that addresses the concerns and suggestions of the reviewers, particularly revising the introduction and discussion sections of the manuscript, and provide additional statistical information on the measures used in this paper.

We look forward to receiving your revised manuscript.

Kind regards,

Sebastian Wallot, Ph.D

Academic Editor

PLOS ONE

Journal Requirements:

3. Please change "female” or "male" to "woman” or "man" as appropriate, when used as a noun (see for instance https://apastyle.apa.org/style-grammar-guidelines/bias-free-language/gender )

Reviewers' comments:

Reviewer's Responses to Questions

**Comments to the Author**

1. Is the manuscript technically sound, and do the data support the conclusions?

Reviewer #1: Yes

Reviewer #2: Yes

2. Has the statistical analysis been performed appropriately and rigorously? 

Reviewer #1: Yes

Reviewer #2: N/A

3. Have the authors made all data underlying the findings in their manuscript fully available?

Reviewer #1: Yes

Reviewer #2: Yes

4. Is the manuscript presented in an intelligible fashion and written in standard English?

Reviewer #1: Yes

Reviewer #2: Yes

5. Review Comments to the Author

Reviewer #1: Dear editor,

I have now reviewed the paper entitled “Lost in the crowd: Imagining walking in synchrony with a crowd increases affiliation and deindividuation”. In the paper, authors asked participants to imagine walking in a crowd in either in coordination or in an uncoordinated manner. They compared the impact of imagined synchronous and non-synchronous movement in a crowd on affiliation and deindividuation. Overall, the paper is well-written and easy to follow. The theoretical background is presented well and the research is grounded on previous work clearly. The methodology followed in the study is capable of answering the research questions stated in the paper. Results and discussion sections are organized comprehensively.

I have few concerns regarding some aspects of the paper:

1. The main starting point of the paper is that in real-life group synchrony might be influenced by cultural primers such as beliefs, ideologies and dressing. In order to control the impact of cultural confounds on group synchrony, authors conducted an online experiment and asked participants to imagine walking in a crowd. In the paper, authors claim that such experimental design would not be confounded by cultural aspects. In the experiment, participants were asked to imagine themselves walking in a crowd. Researchers did not specify a specific crowd. Thus, participants were free to imagine themselves in any kind of group. In one way, this design might considered as a rigorous approach to avoid confounds about group culture. In other way, it can be seen as a limitation: What if the participants considered the groups they are affiliated most when imagining walking in coordination or not? For example, a participant from an army background might have imagined walking with soldiers, a doctor might have imagined walking with other doctors, and so on. I wonder asking participants to imagine walking with “complete strangers” in a crowd would have been a better approach to answer the research questions stated in the paper?

2. In terms of self-reported instruments, authors do not present any statistical findings regarding the validity and/or reliability of those instruments. I am not much familiar with this field. Thus, I wonder if it is OK to ignore the reliability/validity of scales in this line of research?

3. In the Discussion authors claim that “…imagined synchrony interventions can be used to improve the sense of social connectedness for individuals experiencing loneliness and social isolation, and those who are unable to physically coordinate.” I think authors should present some strong arguments/evidence on this practical implication. As far as I know, social relatedness is about seeing one’s self as part of a specific social group (e.g. family, tribe, and team). How can one deal with social loneliness by imagining being in synch with a group that has no cultural relevance to him/her? How can a social system exist without common cultural features? I understand authors point in exploring how synchrony manifests itself when cultural features are somehow wiped out. However, I am not sure the current findings are applicable to the real-life settings where cultural norms and features have inevitable components of groups.

Reviewer #2: The authors investigate whether imaging walking in synchrony with a crowd promotes in the participants feelings of deindividuation and affiliation with the crowd compared to imaging walking asynchronously, as well as whether individuation mediates the effect of imagined synchrony on affiliation.

In contrast with a large part of the studies available in the literature, the study focuses on the embodied elements of deindividuation rather than on the cognitive elements, and it tackles the interesting issue of whether imagined synchrony produces the same effect as physical synchronous actions with respect to feelings of deindividuation and affiliation.

It is the first study to show the positive effect of imagining walking in synchrony on affiliation and, as the authors suggests, it opens paths to future (online) research developments on important topics, including social connectedness and the reduction of prejudicial attitudes. Because of this it can be considered of high relevance and originality. Because of this, I recommend it for publication.

However, some minor revisions might contribute to the quality of the manuscript.

• In the introductory section, the authors go from discussing contemporary research on deindividuation and coordination to the theorizing of McNeill (1995) and Durkheim (1912) to then go back to the contemporary research. I would suggest restructuring the section following a chronological order to facilitate the reading.

• Still in the introductory section, the discussion of what it means for crowds to have “personalities” could also be better organized, as the concerning content of the first page seems to eb partially repeated on page 4 and 5.

• In the discussion, the authors claim that “it is important to point out that even though deindividuation may affect affiliation, and affiliation increases pro-sociality, deindividuation does not necessarily always lead to prosocial outcomes.” As the discussion goes, it seems to be suggested that deindividuation promotes feelings of affiliation toward the in-group while potentially leading to even anti-social behaviors toward the out-group. Thus, the important point would be that deindividuation promotes affiliation toward the in-group but not (necessarily) toward out-groups. This point could be better clarified by the authors.

• The cited paper authored by Herrera (2018) is missing in the references.

6. PLOS authors have the option to publish the peer review history of their article (what does this mean?). If published, this will include your full peer review and any attached files.

Reviewer #1: **Yes: **Muhterem Dindar

Reviewer #2: **Yes: **Flavia Felletti

---

## [Decision Letter · Decision Letter 1]

18 Jun 2021

Lost in the crowd: Imagining walking in synchrony with a crowd increases affiliation and deindividuation

PONE-D-21-02076R1

Dear Dr. Cross,

We’re pleased to inform you that your manuscript has been judged scientifically suitable for publication and will be formally accepted for publication once it meets all outstanding technical requirements.

Kind regards,

Sebastian Wallot, Ph.D

Academic Editor

PLOS ONE

Additional Editor Comments (optional):

Reviewers' comments:

Reviewer's Responses to Questions

**Comments to the Author**

1. If the authors have adequately addressed your comments raised in a previous round of review and you feel that this manuscript is now acceptable for publication, you may indicate that here to bypass the “Comments to the Author” section, enter your conflict of interest statement in the “Confidential to Editor” section, and submit your "Accept" recommendation.

Reviewer #1: All comments have been addressed

Reviewer #2: All comments have been addressed

2. Is the manuscript technically sound, and do the data support the conclusions?

Reviewer #1: Yes

Reviewer #2: Yes

3. Has the statistical analysis been performed appropriately and rigorously? 

Reviewer #1: Yes

Reviewer #2: N/A

4. Have the authors made all data underlying the findings in their manuscript fully available?

Reviewer #1: Yes

Reviewer #2: Yes

5. Is the manuscript presented in an intelligible fashion and written in standard English?

Reviewer #1: Yes

Reviewer #2: Yes

6. Review Comments to the Author

Reviewer #1: Dear editor,

The reviewers have addressed my points sufficiently. I have no further comments. To me, the manuscript can be accepted for publication in PlosONE. I would like to congratulate authors for their interesting work.

Regards,

Reviewer #2: (No Response)

7. PLOS authors have the option to publish the peer review history of their article (what does this mean?). If published, this will include your full peer review and any attached files.

Reviewer #1: **Yes: **Muhterem Dindar

Reviewer #2: **Yes: **Flavia Felletti

---

## [Editor Report · Acceptance letter]

16 Jul 2021

PONE-D-21-02076R1 

Lost in the crowd: Imagining walking in synchrony with a crowd increases affiliation and deindividuation 

Dear Dr. Cross:

I'm pleased to inform you that your manuscript has been deemed suitable for publication in PLOS ONE. Congratulations! Your manuscript is now with our production department. 

Kind regards, 

on behalf of

Dr. Sebastian Wallot 

Academic Editor

PLOS ONE